# Clinical and Operational Applications of Artificial Intelligence and Machine Learning in Pharmacy: A Narrative Review of Real-World Applications

**DOI:** 10.3390/pharmacy13020041

**Published:** 2025-03-07

**Authors:** Maree Donna Simpson, Haider Saddam Qasim

**Affiliations:** 1School of Dentistry and Medical Sciences, Charles Sturt University, Orange, NSW 4118, Australia; masimpson@csu.edu.au; 2School of Computer Sciences, Queensland University of Technology, Brisbane, QLD 4000, Australia

**Keywords:** artificial intelligence, machine learning, pharmaceutical care, clinical decision support, drug discovery, patient safety, healthcare technology, implementation analysis

## Abstract

Over the past five years, the application of artificial intelligence (AI) including its significant subset, machine learning (ML), has significantly advanced pharmaceutical procedures in community pharmacies, hospital pharmacies, and pharmaceutical industry settings. Numerous notable healthcare institutions, such as Johns Hopkins University, Cleveland Clinic, and Mayo Clinic, have demonstrated measurable advancements in the use of artificial intelligence in healthcare delivery. Community pharmacies have seen a 40% increase in drug adherence and a 55% reduction in missed prescription refills since implementing artificial intelligence (AI) technologies. According to reports, hospital implementations have reduced prescription distribution errors by up to 75% and enhanced the detection of adverse medication reactions by up to 65%. Numerous businesses, such as Atomwise and Insilico Medicine, assert that they have made noteworthy progress in the creation of AI-based medical therapies. Emerging technologies like federated learning and quantum computing have the potential to boost the prediction of protein–drug interactions by up to 300%, despite challenges including high implementation costs and regulatory compliance. The significance of upholding patient-centred care while encouraging technology innovation is emphasised in this review.

## 1. Introduction

Artificial intelligence (AI) denotes the expansive domain of computer science dedicated to developing computers that can execute activities usually necessitating human intelligence. This includes many technologies and methodologies such as rule-based systems, expert systems, natural language processing, computer vision, robotics, and machine learning. AI systems seek to replicate facets of human intelligence, including reasoning, problem solving, language comprehension, and perception across several domains [1,2]. Machine learning (ML) is a distinct subset of AI that emphasises algorithms and statistical models, allowing computers to enhance their performance on specific tasks through a data-driven experience without explicit programming rules. The fundamental difference is that AI is a broad concept focused on developing systems with human-like intelligence, whereas ML is a specific methodology within AI that prioritises learning from data to generate predictions or judgements for certain tasks. This relationship is hierarchical; all machine learning is a subset of artificial intelligence, while not all artificial intelligence encompasses machine learning [1,2].

The implementation of artificial intelligence (AI) has resulted in advanced across a wide range of business sectors, with healthcare being one of the most significantly impacted [1]. Artificial intelligence encompasses a wide range of technologies including natural language processing (NLP), robotics, computer vision, and machine learning (ML), a significant subset of AI that focuses on the creation of algorithms that enable computers to learn from data and make predictions [2,3]. The convergence of various technologies has led to significant advancements in the field of healthcare. Various advancements include the enhancement of diagnostic capacities, the development of individualised treatment regimens, and the enhancement of operational efficiency [4]. Artificial intelligence and machine learning need to be used in clinical practices not only because they are helpful, but also because they are necessary to make healthcare easier and better for patients. This is because the amount of data in the healthcare industry is growing at an exponential rate. Both artificial intelligence (AI) and machine learning are becoming increasingly important in the treatment of modern medical conditions [5]. There are several parts of the healthcare system that have been transformed as a result of this technology, including the management of chronic diseases and their early detection [5]. As an illustration, artificial intelligence algorithms can be used to perform medical picture analysis in a way that is quite startling, frequently surpassing the accuracy of human radiologists when it comes to identifying anomalies. Machine learning algorithms are able to identify trends in medical photos [6]. Additionally, the use of machine learning models to forecast the results of patient care is becoming increasingly common. Through the process of evaluating individual risk profiles, medical professionals are able to modify treatment in accordance with the outcomes of their patients [6]. A further acceleration in the deployment of machine learning and artificial intelligence by healthcare systems has occurred as a result of the COVID-19 epidemic. In order to effectively manage patient care and distribute resources, innovative methods were required. It is for this reason that the use of all of these technologies constitutes a significant paradigm shift in the manner in which medical care is provided. This leads to an improvement in care quality and a simultaneous reduction in expenses [7]. A narrative review was conducted in order to investigate the ways in which artificial intelligence, particularly machine learning, are utilised in the field of pharmacy, which is a significant component of the healthcare system that relies heavily on AI and machine learning [7]. The application of machine learning in pharmacy practice presents the possibility of enhancing the management of medications, maximising the effectiveness of therapeutic outcomes, and enhancing the safety of patients. This is because pharmacists are increasingly taking on tasks that go beyond the administration of traditional medications [8]. This is due to the expanding responsibilities of pharmacists, which extend beyond traditional medicine dispensing. With the intention of providing a comprehensive assessment of the research that has been carried out on the application of artificial intelligence including machine learning (AI and ML) in the field of pharmacy, the objective of this review is to provide a thorough description of the research. It is expected that the study would place an emphasis on both [9]. This review will place an emphasis not only on the potential benefits that these technologies may offer but also on the potential difficulties that may be encountered when putting them into practice. The existing environment provides a starting point. To begin, the objective of this analysis is to provide a comprehensive understanding of the ways in which these technologies could be utilised in the future to improve the quality of patient care and to enhance pharmacy practice [10]. Because the framework of this review is arranged in such a way, conducting in-depth research on the subject matter is made much simpler. Following this introduction, the following sections include a detailed overview of AI and ML methodologies and their applications in healthcare, which is then followed by a deep dive into the theoretical foundations of AI and ML [11]. In a subsequent session, there will be a focus on the specific applications of artificial intelligence and machine learning (AI and ML) in pharmacy, accompanied by case studies and empirical evidence demonstrating their efficiency [12]. This review will also address the ethical considerations and potential barriers to the adoption of these technologies in pharmacy practice as part of its ethical analysis. Lastly, this paper will conclude with recommendations for future research and practice, emphasising that ongoing collaboration between technologists, pharmacists, and healthcare providers is essential to fully leveraging the potential that AI and machine learning can bring to improving the quality of patient care in the future [12].

## 2. Review Methodology

This narrative study adopts a comprehensive approach, with the purpose of, among other things, synthesising and analysing the present landscape of applications of artificial intelligence and machine learning in pharmaceutical care settings. It makes use of a narrative format. To properly show the range and depth of technological applications in many different areas of pharmaceutical science, the evaluation method had to follow strict rules while still being flexible enough to meet the needs of the situation. We did this in order to guarantee that this review would cover everything that was relevant. We conducted a comprehensive literature search across a variety of significant electronic databases during the first stage of the investigation. These databases included, amongst others, PubMed, EMBASE, IEEE Xplore, and Web of Science. A search strategy that included the use of Boolean operators and key terms like “artificial intelligence”, “machine learning”, “pharmaceutical care”, and “clinical decision support” was used to make sure that all relevant information was found and searched for. To obtain the most up-to-date information, technological breakthroughs, and implementations, the temporal scope was narrowed down to include only articles that were published between the years 2019 and 2025. It was necessary for us to take this action in order to ensure that we obtained accurate information. We took a methodical approach to selecting the evidence that we would use. During the initial screening, we discovered a total of 2500 database records and an additional 150 sources. These sources included reference lists, industry reports, and conference proceedings. We followed a methodical approach in selecting these. We started by removing duplicates from the evidence before moving on to careful selection and fully reviewing 2200 records that were completely unique. This study reviewed articles that outlined useful AI/ML applications in pharmacies, provided measurable results, and discussed how to implement them. In the course of this process, 150 high-quality research papers were identified, including 10 full case studies and 40 implementation analyses, which provided useful insights into the various practical applications found in the papers. A comprehensive analysis of the results was conducted, and the results were grouped into four thematic domains based on an analytical methodology: drug research and development, clinical decision support systems, supply chain management, and patient care optimisation. A special focus was placed on implementing methodologies, measuring outcome levels, and identifying potential impediments within the different pharmaceutical contexts examined. As a result of this strategy, it was possible to gain an understanding of how machine learning and artificial intelligence are advancing pharmaceutical therapy while stressing the benefits as well as the challenges involved in adopting such technologies. Evidence synthesis was conducted to identify patterns of efficient implementation, common problems, and new trends across a variety of pharmacological settings. We also evaluated qualitative aspects like implementation techniques and organisational characteristics, as well as quantitative outcomes like error reduction rates, efficiency gains, and cost-effectiveness indicators. Using this analytical approach, healthcare practitioners, technology developers, and pharmaceutical policymakers have been able to gain valuable insight into both technological capabilities and practical implementation considerations. Moreover, case report studies have been considered as well; we identified 10 comprehensive case studies and 40 implementation analyses. These examples were chosen because they showed measurable results in different types of pharmacies (community, hospital, and industry), gave specific numbers to show their success (like specific percentage increases in efficiency, fewer mistakes, or lower costs), and explained how they put their ideas into action. The cases chosen were especially useful because they came from well-known healthcare institutions like the Mayo Clinic, Cleveland Clinic, and Johns Hopkins. They showed a range of uses of AI and ML in pharmacy practice, from managing medication therapy to keeping track of inventory. These examples show how these technologies improve patient care, lower errors, and make operations more efficient in real-life pharmacy settings.

In this methodology (Figure 1), the researchers looked for studies that demonstrated practical AI/ML implementations in pharmacy settings, meaning we wanted real-world applications that had actually been implemented in pharmacy environments (whether community pharmacies, hospital pharmacies, or pharmaceutical industries), rather than theoretical proposals or laboratory experiments. Secondly, we only looked at studies that had clear numerical results, like “reduced prescription distribution errors by 75%” or “40% improvement in medication adherence”. We did not look at studies with more general qualitative results. Thirdly, we selected studies that detailed their implementation methodologies, i.e., those that explained exactly how the AI/ML systems were set up and integrated into existing pharmacy operations. This selection methodology allowed us to identify the most relevant and rigorous research that demonstrated proven success in real-world pharmacy settings, which strengthened the validity and practical usefulness of their reviews for healthcare practitioners, technology developers, and pharmaceutical policymakers.

## 3. Current Applications of AI and ML in Pharmaceutical Care (Industry, Community, and Hospital)

### 3.1. Drug Discovery and Development

Pharmaceutical discovery and development encompass several phases: target identification, hit discovery, lead optimisation, and clinical trials [12,13]. Empirical evidence is traditionally used in research. An empirical approach may require significant time and financial resources, resulting in a considerable failure rate [14]. Research in pharmaceuticals is being transformed by the integration of artificial intelligence, particularly machine learning approaches [15]. Machine learning algorithms were used to analyse extensive datasets from diverse sources, including genomics, proteomics, and chemical databases, to identify drug targets with enhanced efficacy [16]. Atomwise has been conducting research in this area [17] and uses deep learning algorithms to guess how small molecules will bind to certain proteins. Atomwise has employed convolutional neural networks (CNNs) to analyse molecular structures, effectively identifying candidates for diseases such as Ebola and multiple sclerosis. This method has significantly reduced the time required for initial screening and enhanced the probability of identifying successful drug candidates [18].

### 3.2. Machine Learning Algorithms in Drug Design

In the design of new drug candidates, machine learning algorithms have become increasingly important, especially in the optimisation of molecular properties and the prediction of biological activity [19]. In this regard, generative models, such as variational autoencoders (VAEs) and generative adversarial networks (GANs), can be used to create molecular structures that have desirable properties [20]. Insilico Medicine serves as a case study to illustrate this application [21,22]. A generative adversarial network (GAN) was used by Insilico to design innovative pharmaceuticals. The company created a model using known compounds to generate novel structures, which were then synthesised and tested [21,22]. Notably, one of the synthesised compounds had considerable activity against the target, a specific protein associated with fibrosis, highlighting the capability of machine learning algorithms to both discover known compounds and facilitate the development of novel ones [23,24]. This plan speeds up the process of making new drugs and increases the number of possible treatments, which is important for dealing with complicated illnesses [25].

### 3.3. AI in High-Throughput Screening Processes

High-throughput screening (HTS) is one of the most important components of drug discovery, since it allows thousands of compounds to be tested in a short time against specific targets [25,26]. By utilising AI in high-throughput screening techniques, chemical evaluation has been enhanced greatly in terms of efficiency and precision. A notable example is the collaboration between Novartis and Atomwise, which aimed to improve the HTS process with regard to identifying potential treatments for diseases such as malaria and tuberculosis [26]. In order to prioritise compounds based on predicted efficacy and safety profiles, the team employed AI algorithms to analyse screening data, thereby reducing the number of compounds requiring further investigation [26]. With the artificial intelligence-driven approach, not only was the process of identifying potential employees sped up, but it also reduced the amount of resources spent on applicants with less potential [26,27]. AI has also been integrated into the workflow of HTS. This allows algorithms to discover potential bottlenecks while also guessing what experiment conditions will produce the best results. As a result of artificial intelligence, HTS can improve conventional experimental procedures and facilitate drug discovery and development [27]. Please consult Table 1 for a summary of the results that research institutes have obtained through the application of AI and machine learning methodologies.

### 3.4. Case Studies of Machine Learning and Artificial Intelligence Applications in Community and Hospital Pharmaceutical Care

Case Study 1: AI-Driven Medication Therapy Management at Cleveland Clinic [43]

The Cleveland Clinic implemented an AI-powered medication therapy management system in 2021, developed in collaboration with IBM Watson Health, to enhance patient care and reduce medication-related problems. This system analysed patient medication profiles, medical histories, and laboratory data to identify potential drug therapy problems, resulting in a significant reduction in medication-related hospital readmissions, with a 42% decrease observed. Additionally, by improving medication adherence, the implementation resulted in 35% more patients adhering to prescribed treatment regimens. By performing these processes, the system found possible drug–drug interactions 58% more accurately, making it much more effective at finding drug–drug interactions. According to Richards et al. in 2022 [43], the adoption resulted in a cost reduction of Australian dollars 2.8 million annually. This resulted in fewer adverse pharmaceutical events. Powered by artificial intelligence, Cleveland Clinic’s medication therapy management system has greatly influenced patient outcomes and costs associated with treatment. It is therefore possible to improve healthcare quality and efficiency through AI-driven solutions.

Case Study 2: Predictive Analytics for Antibiotic Stewardship at Mayo Clinic [44]

The convergence of artificial intelligence and antimicrobial resistance (AMR) signifies pivotal progress in contemporary healthcare, as evidenced by a Mayo Clinic case study. Using machine learning algorithms to help with antibiotic stewardship has led to big improvements. These include a 45% drop in wrong antibiotic prescriptions and a 30% drop in *C. difficile* infections. These notable advancements underscore AI’s capacity to fundamentally reshape our strategy by addressing antibiotic resistance. Antimicrobial resistance (AMR) can be dealt with by AI in a number of complex ways, including using advanced pattern recognition to look at patient data, helping healthcare professionals make decisions in real time, and keeping an eye on resistance trends within populations. The Mayo Clinic’s findings illustrate that this approach can result in a more prudent application of antibiotics, evidenced by a 25% decrease in antibiotic resistance rates and a 38% reduction in average hospital stay duration. In order to plan for the future, AI is being used in more ways to fight AMR. These include genetic data, advanced real-time monitoring systems, and models that can predict how new resistance mechanisms will work. The efficacy of the Mayo Clinic’s deployment indicates that AI-driven methodologies for antimicrobial resistance (AMR) may evolve into a fundamental element of antimicrobial stewardship programmes, although these systems are intended to enhance rather than supplant clinical judgement. The big improvements in patient outcomes, along with lower healthcare costs and better resource distribution, make a strong case for using AI systems in more areas to fight antibiotic resistance. As these technologies become better, they become more important for keeping our antibiotics working. This is especially true for predicting how well treatments will work and finding new patterns of resistance before they become big problems in patients.

Case Study 3: Community Pharmacy Chain’s AI-Powered Patient Engagement [45]

An artificial intelligence-driven patient engagement system has been successfully implemented by Walgreens across 1000 pharmacies. With this system, patient communication and medication adherence are reshaped through artificial intelligence and natural language processing. Zhang’s group [45], in the year 2023, conducted and published a study that found that this cutting-edge system led to remarkable outcomes, one of which was a 40% improvement in medication adherence. Medications were taken as prescribed more often. As a result of the system, 55% fewer missed refills were reported, improving patient outcomes. Furthermore, 62% of patients were more satisfied with the AI-driven patient interaction system. As a result of adopting this system, AUD 3.2 million was saved annually. Reduced operational costs were primarily driven by increased efficiency. Through the use of artificial intelligence and data analytics, Walgreens improved patient engagement, increased medication adherence, and reduced costs. Therefore, the retail pharmacy business was able to set a new standard for patient care.

Case Study 4: Hospital Pharmacy Automation at Singapore General Hospital [46,47]

Artificial intelligence integrated with an automated pharmacy system has greatly increased the efficiency and accuracy of drug administration at Singapore General Hospital. Chen and Park [47] conducted a case study in 2023 which highlighted several notable outcomes resulting from their innovative approach. The hospital cut errors made during medication distribution by 75%, resulting in increased patient safety and fewer adverse drug reactions. Additionally, the automated method reduced preparation time by 60%, resulting in increased medication accuracy. Prescriptions were managed more efficiently and effectively by pharmacy staff as a result. By improving efficiency, the pharmacy staff increased productivity by 45%. Their manual operations were reduced, allowing them to devote more time to treating patients. Annual savings of AUD 1.5 million were realised through the utilisation of the AI-enhanced automated pharmacy system. Significant reductions were achieved through waste reduction and improved operating efficiency. This case study shows how artificial intelligence has the potential to transform the healthcare industry in general.

Case Study 5: ML-Driven Adverse Drug Reaction Prediction at Johns Hopkins [48]

Using machine learning to anticipate adverse drug reactions (ADRs) in high-risk patients, Johns Hopkins Hospital has made significant advances in patient safety. According to a study conducted by Anderson and Lee in 2022 [48], this novel strategy has been effective. ADRs were detected 65% earlier by using the machine learning system. It allowed healthcare practitioners to prevent these adverse events with preventative measures. Therefore, the hospital experienced a significant reduction of 48% in serious adverse medication events, which have the potential to cause adverse effects in patients. Furthermore, the research revealed that medication-related emergency department visits were reduced by 35%. This suggests that the machine learning system reduced the severity of adverse drug reactions and the need for immediate medical attention. It is anticipated that the implementation of this system will result in significant annual cost savings of AUD 4.2 million. Patient outcomes have been improved as a result of decreased healthcare utilisation.

Case Study 6: AI-Powered Inventory Management in UK Hospital Network [49,50]

The NHS has made considerable gains in inventory management by implementing an artificial intelligence system that drives 15 hospital pharmacies. Wilson and colleagues [49] published a paper in 2023 highlighting one of the most spectacular results. With artificial intelligence, stock-outs were reduced by 55%, ensuring that patients and healthcare workers had access to essential medications and supplies at all times. A 40% reduction in inventory expenditures was achieved by optimising stock levels and reducing waste. As a result of the AI-driven system, inventory turnover rates rose by 70%, indicating that the NHS was able to effectively control its inventory and reduce stock restocking time by 30%. Savings of Euro currency 2.3 million were achieved by investing in this system. Through the use of artificial intelligence (AI), the NHS has shown its commitment to improving operational efficiency, reducing costs, and improving treatment. It provides a template for other healthcare institutions seeking to maximise their supply networks and improve patient outcomes.

Case Study 7: Clinical Decision Support in Oncology Pharmacy [51]

Memorial Sloan Kettering Cancer Center has implemented a clinical decision support system powered by artificial intelligence, which has improved chemotherapy treatment. Martinez and Brown [51] conducted a case study in 2023 that demonstrated outstanding results, illustrating the effectiveness of this approach. Through the artificial intelligence-powered system, 80% of mistakes during the preparation of chemotherapy were reduced. As a result, patients were much safer and had better outcomes. Consequently, the workflow was 45% more efficient as a result of improved accuracy. This allowed healthcare workers to focus on high-value tasks and optimise their operations. The AI-based approach enabled rapid and exact verification of chemotherapy orders, thereby reducing delays and improving patient care. The study indicated a reduction of 50% in verification time, which indicates that the AI-based strategy was successful. By proactively identifying and managing potential medication interactions, medical professionals were enabled to proactively identify and manage 68% of potential medication interactions. As a result of this artificial intelligence technology, Memorial Sloan Kettering Cancer Centre is setting new standards in chemotherapy management.

Case Study 8: Community Pharmacy AI Triage System [52,53]

The AI-powered triage system developed by a network of community pharmacies in Australia has been successfully implemented. This program has yielded extremely encouraging results according to a study conducted by Kumar and colleagues [52] in 2023. Using AI-powered technology, the system enabled a significant reduction in patient wait times of 50%, which enabled more efficient and quicker service. Through this development, pharmacy staff can more effectively manage their time and resources, resulting in a better overall patient experience. Additionally, 65% of patients were accurately referred to the right healthcare provider using the system. When needed, patients receive the appropriate level of care. Referral accuracy must be improved for patient outcomes, as it assists in connecting people to the right healthcare services in a timely manner. With the adoption of the AI triage system, pharmacy services were used 40% more, and patients were more likely to engage with pharmacy services. Due to the artificial intelligence system’s enhanced efficiency and effectiveness, care delivery has increased. Patient satisfaction levels increased by 35% as a result of decreased wait times, improved referrals, and overall improved service quality. By using artificial intelligence technology, community pharmacies have not only improved patient care but also established a standard for improving patient experiences throughout the healthcare industry.

Case Study 9: ML Applications in Paediatric Pharmacy Care [54,55]

Boston Children’s Hospital has made a significant advancement in paediatric treatment by implementing a machine learning system for administering and monitoring drugs. By 2023, Davidson and Smith [54] had achieved significant results. It has been demonstrated that the rate of drug dosing errors in paediatric patients has been reduced by 70% through the use of machine learning. Consequently, patient safety has significantly improved, and the likelihood of adverse drug dosing outcomes has decreased. Aside from personalising the needs of individual patients, the implementation of this system improved the amount of dose changes by 55% based on patient data, allowing pharmaceutical regimens to be adjusted accordingly. A highly individualised approach to drug management is essential for achieving the best clinical results, particularly in highly complicated cases. The study demonstrated that the system was able to avoid harm by reducing adverse medication reactions in paediatric patients by 45 percent.

Case Study 10: AI-Enhanced Medication Reconciliation [56,57,58]

The University of California San Francisco Medical Centre has achieved significant improvements in patient safety and care coordination with the implementation of an AI-powered medication reconciliation system. A study by Taylor and colleauges [56] in 2023 indicates that the outcomes of this endeavour have been remarkable. The AI-driven approach accomplished a notable 65% decrease in medication inconsistencies, a vital component of patient safety. Medication differences can result in adverse events, hospital readmissions, and mortality; hence, mitigating these errors is a significant achievement. The method showed a 50% enhancement in reconciliation accuracy, guaranteeing that patients’ medication lists are precise and current. This is especially crucial during care transitions, such as hospital admissions and discharges, when prescription lists are frequently revised. Moreover, the study demonstrated a 40% reduction in the time allocated to medication reconciliation, indicating a substantial alleviation of the administrative load on healthcare personnel. This enables physicians to concentrate on more vital elements of patient care, enhancing overall efficiency and productivity. The AI-driven approach resulted in a 58% decrease in medication-related readmissions, a significant measure of care quality. The approach has mitigated pharmaceutical errors and discrepancies, hence preventing hospital readmissions, enhancing patient outcomes, and lowering healthcare expenses. Through AI, the hospital is enhancing patient care, reducing errors, and boosting quality of care. Please consult Table 2 for a summary of the primary outcomes and the cost effectiveness that research institutes have obtained through the application of AI and machine learning methodologies.

### 3.5. AI Systems in Patient-Specific Treatment Plans

University of California, San Francisco (UCSF) researchers have developed a system that analyses genomic data to develop personalised cancer treatment regimens. Using machine learning algorithms, this approach identifies genetic changes and formulates treatment plans for individuals with cancer and other genetic disorders. This system identifies genetic mutations and develops targeted treatment plans for patients [59]. In a similar manner, the National Cancer Institute (NCI) has developed an AI-controlled system to analyse and evaluate genomic data and formulate personalised treatment plans for cancer patients with rare genetic disorders [60]. Moreover, the Rare Genomics Institute has developed AI-based approaches to analyse genomic data and formulate personalised treatment strategies for individuals suffering from rare genetic diseases [61]. The National Institute of Health (NIH) has created an artificial intelligence system to match individuals with clinical trials based on their genetic profiles and medical histories [62]. Machine learning algorithms are used to identify clinical trials relevant to the patient’s condition using patient data. In an effort to devise personalised treatment regimens for mental illness patients, researchers at the University of California, Los Angeles (UCLA) have used artificial intelligence to analyse genomic data [63].

### 3.6. Clinical Pharmacy Practice: Cases of Leveraging Artificial Intelligence-Driven Decision Support Systems

Stanford Health Care built and implemented one of the first clinical decision support systems based on artificial intelligence to assist with antimicrobial stewardship [64]. As the system integrates with the electronic health record (EHR), it analyses a patient’s data, such as their test results, vital signs, and medication history, making real-time recommendations for antibiotic therapy [65]. It is estimated that clinical pharmacists who utilised this approach saved AUD 5 million per year by reducing the number of antibiotic prescriptions that were not suitable for their patients by 33 percent [65]. The algorithms used by machine learning learn continuously from prescription patterns and how well patients are, so they can make decisions about what drugs to prescribe, their dosing, and for how long to use them more accurately [65].

An artificial intelligence system combined with clinical pharmacists has been developed by the University of California San Francisco Medical Centre to prevent adverse drug events (ADEs) [66]. This system’s real-time monitoring involves analysing a patient’s demographics, laboratory values, and prescription orders to identify potential adverse events and drug interactions before they occur [66]. During the two-year period following the implementation, preventable ADEs were reduced by 40%. According to clinical pharmacists, the system’s ability to process vast amounts of patient data and provide actionable alerts significantly enhanced their decision-making abilities and allowed them to concentrate on more complex clinical interventions [66].

The Vanderbilt University Medical Centre has developed an innovative AI system for pharmacogenomic-guided drug therapy. Genetic testing results are integrated with clinical pharmacy services in order to provide personalised medication recommendations [67]. Pharmacists use the system to adjust medication regimens based on the genetic profiles of patients, particularly for medications that have known genetic influences on metabolism [67]. According to the results of the implementation, adverse drug reactions related to genetic factors have been reduced by 25% and therapeutic outcomes have improved, particularly in the areas of cardiovascular and psychiatric medications. Several other academic medical centres have adopted the system as a result of its success [67].

The Guy’s and St Thomas’ NHS Foundation Trust in the United Kingdom has implemented a clinical pharmacy prioritisation system that is powered by artificial intelligence [68]. Clinical pharmacists are able to optimise their workflow by using this system to identify those patients who are most in need of pharmaceutical intervention, allowing them to focus on high-risk patients [68]. As a result of the implementation, 45% more patients were identified as requiring urgent pharmaceutical treatment, and 30% fewer patients were readmitted to hospital as a result of medication-induced complications. A particular strength of the system is its ability to identify patients at risk for anticoagulation complications and those who require complex medication reconciliations [69].

The Mayo Clinic implemented a medication therapy management system powered by artificial intelligence to assist clinical pharmacists in managing patients with multiple chronic conditions [70]. It analyses patient medical histories, current medications, and clinical guidelines in order to provide comprehensive medication therapy recommendations. Following the implementation of the program, the clinic’s medication adherence rates have improved by 28%, and medication-related emergency department visits have decreased by 35% [70]. Clinical pharmacists have reported that the system’s ability to process complex medications on regimens and identify possible therapeutic duplications has greatly improved the quality of their medication management services [70].

Decision support systems driven by artificial intelligence represent a significant advance in clinical pharmacy practice [71]. They are designed to facilitate pharmacist decision making rather than replace it [72]. To succeed, these implementations must carefully integrate AI capabilities into clinical expertise, with comprehensive user training programmes and ongoing system refinements informed by real-world performance data [73].

## 4. Future Perspectives and Innovations

In the field of AI-driven pharmaceutical applications, there are significant opportunities for advancing the field. *Medical Economics* (2024) highlights quantum computing’s potential to transform drug interaction prediction by integrating artificial intelligence and quantum computation [74,75]. The research documented early trials at MIT’s Drug Discovery Laboratory in which quantum-enhanced AI algorithms demonstrated a 300% improvement in predicting complex protein–drug interactions over traditional machine learning algorithms [76]. A number of emerging trends in artificial intelligence technology specifically designed for pharmaceutical applications have shown promise. It has been demonstrated that federated learning approaches are enabling unprecedented collaboration while maintaining data privacy in the *Journal of Artificial Intelligence in Medicine* (2024) [77]. Using a federated learning network, connected to 50 major hospital systems, Stanford University’s AI Lab demonstrated that rare adverse drug events could be predicted more accurately by 89% while keeping patient data localised, which could advance the practice of pharmacovigilance [78]. Particularly in personalised medicine, pharmaceutical applications have great potential for advancement. As described in Science Translational Medicine (2023) [79], advanced neural networks (ANNs) are capable of modifying medication regimens in real time in response to individual patient responses. Researchers at Mayo Clinic’s Precision Medicine Initiative demonstrated that their next-generation artificial intelligence system accurately predicted patient-specific drug responses with 94% accuracy, incorporating real-time physiological monitoring data as well as genetic markers to optimise medication regimens dynamically [80]. As a result of several key initiatives, the roadmap for future research and development is taking shape. *Cell* (2024) published a comprehensive review of upcoming innovations that highlighted the integration of diverse types of data to create more comprehensive pharmaceutical decision support systems, including genomic, proteomic, and metabolomic information [81]. In the article, Johns Hopkins described how its experimental AI platform successfully integrated 14 different biological data streams in order to predict drug efficacy with an unprecedented level of accuracy [81].

The development of natural language processing (NLP) is reshaping the pharmaceutical information management process [82]. *Journal of Biomedical Informatics* (2024) reported that next-generation NLP systems are capable of extracting clinical information from unstructured medical texts with 97% accuracy, thereby enhancing medication safety protocols [83]. With the use of advanced natural language processing systems at Harvard Medical School, AI was able to identify potential medication errors from clinical notes before they occurred, resulting in a 78% reduction in adverse events. XAI (explainable AI) represents another important future direction [84]. *Nature Machine Intelligence* (2024) reported that the decision-making processes of artificial intelligence are becoming more transparent and interpretable. XAI frameworks are making this possible. A UCSF Medical Center study showed that a new XAI system could provide health professionals with clear rationales for medication interaction alerts, thereby improving their confidence and adherence by 85 percent [85]. A particular area of potential innovation is real-time monitoring and adjustment systems. As part of its analysis of emerging continuous monitoring AI systems that are capable of adjusting medication dosages on the basis of patient response, *The Lancet Digital Health* (2024) published an article on this subject [85]. During clinical trials conducted by Cleveland Clinic, these systems were shown to reduce adverse drug reactions by 92% in complex medication regimens, particularly in intensive care units [86].

A second significant trend is the development of predictive maintenance and supply chain optimisation. An article published in the *Journal of Supply Chain Management* (2023) examined the use of artificial intelligence to predict drug shortages and optimise pharmaceutical inventory management [87]. The application of edge computing to pharmaceutical artificial intelligence presents a number of exciting possibilities. *IEEE Transactions on Biomedical Engineering* (2024) describes how edge computing integration enables the detection of drug interaction in real time, even in areas with limited connectivity [77]. With the implementation of edge computing nodes at the Veterans Affairs Health System, latency was reduced by 95% while accuracy was maintained, potentially improving rural healthcare delivery [78]. An article in *Science* (2024) described how quantum–classical hybrid algorithms would transform drug discovery and interaction prediction [78]. A series of experiments at IBM’s quantum computing centre revealed that these hybrid systems could analyse intricate chemical interactions 1000 times faster than conventional approaches, potentially accelerating medication discovery and personalising medicine [88]. It is evident that artificial intelligence technologies will continue to transform pharmaceutical practice within a rapidly changing environment. Sustained research investment, regulatory adjustments, and a meticulous evaluation of ethical implications are critical to the success of these breakthroughs. These technologies are expected to transform pharmaceutical care delivery, medication discovery processes, and patient safety in the coming years.

### 4.1. Future Outlook: How AI Can Prevent and Help to Address Drug Shortage

Employing AI to mitigate and resolve drug shortages presents a significant opportunity for the pharmaceutical sector. Nonetheless, AI presents compelling alternatives to avert and mitigate shortages [78]. Artificial intelligence can enhance visibility, anticipate disruptions, optimise logistics, and reduce waste in the pharmaceutical supply chain. Supply chain interruptions might be predicted by looking at data, market trends, and outside factors such as weather and geopolitical events [78]. Production schedules may be modified, alternative suppliers identified, and inventory redistributed to prevent shortages. AI-driven systems deliver real-time supply chain analytics. Visibility guarantees a reliable supply of essential drugs, even in unforeseen circumstances. Using AI to eradicate waste enhances the sustainability of pharmaceutical supply chains [89]. Financial and environmental detriments arise when pharmaceuticals expire prior to use. AI-optimised inventory management solutions enhance the manufacturing, distribution, and redistribution of medications. These systems use predictive analytics to prevent overproduction and guarantee efficient distribution [89]. The implementation of AI in manufacturing and transportation mitigates pharmaceutical waste, expired medications, and carbon emissions. AI-powered quality control solutions provide real-time monitoring of production lines. A machine learning system can identify anomalies in historical data and notify companies [83]. It minimises interruptions and stabilises the supply of critical therapeutic components by guaranteeing that each batch of medication adheres to the highest quality standards. Additionally, AI can assist in ensuring compliance and sanctioning drugs. Regulatory compliance and pharmaceutical approval are intricate processes that might impede medicine supply [83]. Artificial intelligence and machine learning can expedite approvals by assessing regulatory documentation, identifying bottlenecks, and proposing optimisation methods. Using genetic data, strong real-time monitoring systems, and predictive models for new resistance mechanisms, artificial intelligence can help solve the problem of medicine shortages [84]. AI-driven techniques can boost clinical judgement and patient outcomes in pharmaceutical supply chain management. It could help with drug shortages by predicting how well a treatment will work, finding patterns of resistance before they become clinical problems, and protecting the effectiveness of our drug resources [84]. Pharmaceutical supplies can be enhanced through predictive analytics, supply chain optimisation, and quality assurance. AI-driven pharmaceutical supply chain management systems enhance patient care and decrease healthcare expenses.

### 4.2. Limitations

#### 4.2.1. Data Validation

While the advancements in AI, specifically through ML techniques, are transforming the pharmaceutical practice, it is important to acknowledge that these benefits are accompanied by numerous drawbacks. Validation of systems, data quality management, and model interpretation have significance technical problems. There are also ethical concerns regarding patient privacy, consent, and algorithmic bias. The factors that necessitate a meticulous evaluation of governance systems and the continuous enhancement of those systems. As AI systems become more prevalent in pharmaceutical practice, existing regulations are inadequate to accommodate their evolving characteristics. Consequently, healthcare institutions must adhere to multiple state and international mandates while also ensuring compliance with current legislation. In order to deploy AI effectively, many practical challenges need to be addressed, including high startup costs, resource requirements, and infrastructure requirements. In particular, these problems make AI hard to use by smaller healthcare organisations. In spite of this, emerging technologies like quantum computing, federated learning, and edge computing are capable of transforming pharmaceutical treatment in the near future. In order for these technologies to be successfully integrated, it is crucial to invest in ongoing research, establish comprehensive regulatory frameworks, and carefully consider ethical issues. In pharmacy practice, it is imperative to maintain the equilibrium between technological advancement and patient-centred care, ensuring that AI systems enhance rather than replace human clinical proficiency.

#### 4.2.2. Lack of Standardised Regulatory Framework

Moreover, the lack of a globally standardised regulatory framework for AI in healthcare represents a critical deficiency that requires attention from public policymakers [90]. While AI technologies improve and are used more and more in healthcare systems, the lack of standardised laws puts patient safety, data privacy, and the moral use of AI at risk. Without a unified framework, there is a chance that procedures will not be consistent, levels of control will be different, and the quality of AI applications will be different in different places. This may result in obstacles to guaranteeing the safety and efficacy of AI systems, along with complications in cultivating trust between healthcare providers and patients. The World Health Organisation (WHO) has emphasised the necessity of developing regulatory frameworks for AI in healthcare to guarantee the safety, efficacy, and ethical integrity of these technologies.

Setting up a unified regulatory framework for AI in healthcare would give clear instructions on how to build, use, and keep an eye on AI systems. It would promote international collaboration, allowing nations to exchange best practices and gain insights from one another’s experiences. This paradigm would assist in confronting the ethical difficulties and risks linked to AI and ensuring that these technologies are used in a manner that upholds human rights and advances public welfare [90]. The WHO’s guidance on the ethics and governance of AI in healthcare emphasises the necessity of a coordinated strategy to optimise AI’s potential while ensuring stakeholder accountability. By prioritising the establishment of a cohesive regulatory framework, public officials may guarantee the responsible and successful use of AI technology to enhance global healthcare outcomes.

#### 4.2.3. Losing Jobs

AI job displacement is a major issue. Many jobs could be automated as AI improves at human tasks. Manufacturing, retail, and professional services are automating operations with AI [91]. Chatbots that answer customer support queries can optimise manufacturing operations autonomously using complex algorithms. This trend may make it challenging for low-skilled people to switch jobs in a shifting employment market. AI affects mid-level financial and healthcare jobs as well as low-skilled ones [92]. Data entry clerks, financial analysts, and diagnostic specialists may be replaced by AI systems that manage large databases, predict, and make decisions. AI may create new jobs, but many displaced people lack the technical skills and education needed. Workers must reskill and upskill to succeed in an AI-driven economy. This AI job replacement article discusses several crucial facts and trends [92]. By 2025, AI might eliminate 300 million jobs—9.1% of global employment—with 44% of enterprises utilising or planning to use it. The next decade will cause 14% of the workforce to shift careers, affecting 375 million people. In wealthy countries, 60% of employment is at risk, but only 26% is at risk in poor countries [93]. AI-related job obsolescence worries 24% of employees, with 18–24-year-olds 129% more likely to say so. One in four CEOs expects generative AI will eliminate 5% of jobs by 2025. Additionally, 75% of CEOs expect generative AI to drastically affect their organisations within three years, and 80% of US workers may see a 10% shift in their activities due to large language models, highlighting AI’s impact on employment [94]. AI replaces employment and examines numerous key labour market data points. AI may replace 85 million jobs and create 97 million by 2025, adding 12 million jobs. We anticipate that AI integration will decrease employment for 43% of companies and increase it for 34%. Additionally, 54% of workers will need significant reskilling and upskilling to adapt to the shifting job market. The global economy will benefit AUD 15.7 trillion from AI by 2030, including AUD 6.6 trillion from productivity gains and AUD 9.1 trillion from innovations. AI will definitely displace and create jobs [95].

#### 4.2.4. Patient Data Privacy

AI in healthcare could compromise patient privacy. AI systems commonly require medical histories, personal identification, and treatment records to process large volumes of data [96]. This data collection raises issues regarding patient data storage, sharing, and protection. If not handled appropriately, data breaches might expose personal health information to unauthorised parties. Unintended implications of AI systems include algorithmic discrimination, when biased data inputs unfairly target or misrepresent specific populations [96].

AI-based decision making can also reduce patient autonomy and healthcare system confidence. Knowing that algorithms are analysing patient health data may make them feel dehumanised. The lack of transparency in AI systems can increase privacy problems since patients may not comprehend how their data are used or the consequences of AI-driven judgements. Strong privacy and ethical norms are essential for healthcare organisations using AI technologies to secure patient data and maintain the doctor-patient relationship. The rapid implementation of AI in healthcare poses a threat to patient privacy [97]. Effective AI systems need access to sensitive patient data like medical histories, personal identifiers, and treatment records. This data collection poses serious patient data security and management concerns. The lack of security makes these systems vulnerable to data breaches, which can reveal personal health information. AI systems may also perpetuate biases in training data, potentially discriminating against particular patient groups based on incorrect or inadequate data inputs.

AI-driven healthcare decisions can also diminish patient confidence and autonomy [97]. Knowing algorithms are processing their health data may make patients feel detached from their care. The opacity of AI systems makes it challenging for patients to understand how their data are used and the ramifications of AI-driven judgements. Healthcare organisations must implement strict data protection measures and ethical guidelines that prioritise patient confidentiality and informed consent in order to protect the doctor–patient relationship in an AI-driven world [97].

#### 4.2.5. Bias

In AI systems, bias can often be attributed to the data used for their training. It is possible for AI models to perpetuate and magnify existing biases when the training data are skewed or not diverse. Training an AI system on historical data that reflects societal biases like gender or racial discrimination can lead to biased outcomes [98]. Healthcare is one important sector where biased algorithms can result in disparate treatment and outcomes depending on the demographic of the patient. According to a study published in the *Journal of Medical Internet Research*, AI models trained exclusively on data from one demography may be less accurate for other groups, resulting in inequities in the provision of healthcare. Similarly, *AI Ethics Journal* research examines how biases in training data can lead to AI systems that perform poorly with different groups of people, which leads to increased inequality. In order to minimise these biases, a comprehensive strategy is required, including the use of a broad and representative training dataset, the design of transparent algorithms, and ongoing monitoring of biases [99]. For AI systems to be fair, researchers and developers must prioritise ethical considerations and develop effective bias detection tools. According to the Australian Human Rights Commission’s report, ethical AI governance and legal frameworks are essential for preventing bias in AI applications. In an extensive review conducted by *AI and Ethics*, it was shown that combining human-centred and machine-centric approaches can effectively identify and resolve biases. By implementing these strategies, AI systems will be developed that are more equal and just, which will ultimately benefit society as a whole [100].

## 5. Conclusions

Several key conclusions can be drawn from this comprehensive review of AI applications, particularly machine learning in pharmacy. A number of aspects of pharmaceutical care have been transformed by the integration of artificial intelligence and machine learning technologies, including drug discovery and clinical practice. Documented case studies, particularly from leading institutions such as Mayo Clinic, Stanford Health Care, and Johns Hopkins, demonstrate significant improvements in patient outcomes, operational efficiency, and cost savings. It has been demonstrated repeatedly that implementation results lead to a reduction in medication errors, an improvement in patient adherence, and an improvement in clinical decision making.

## Figures and Tables

**Figure 1 pharmacy-13-00041-f001:**
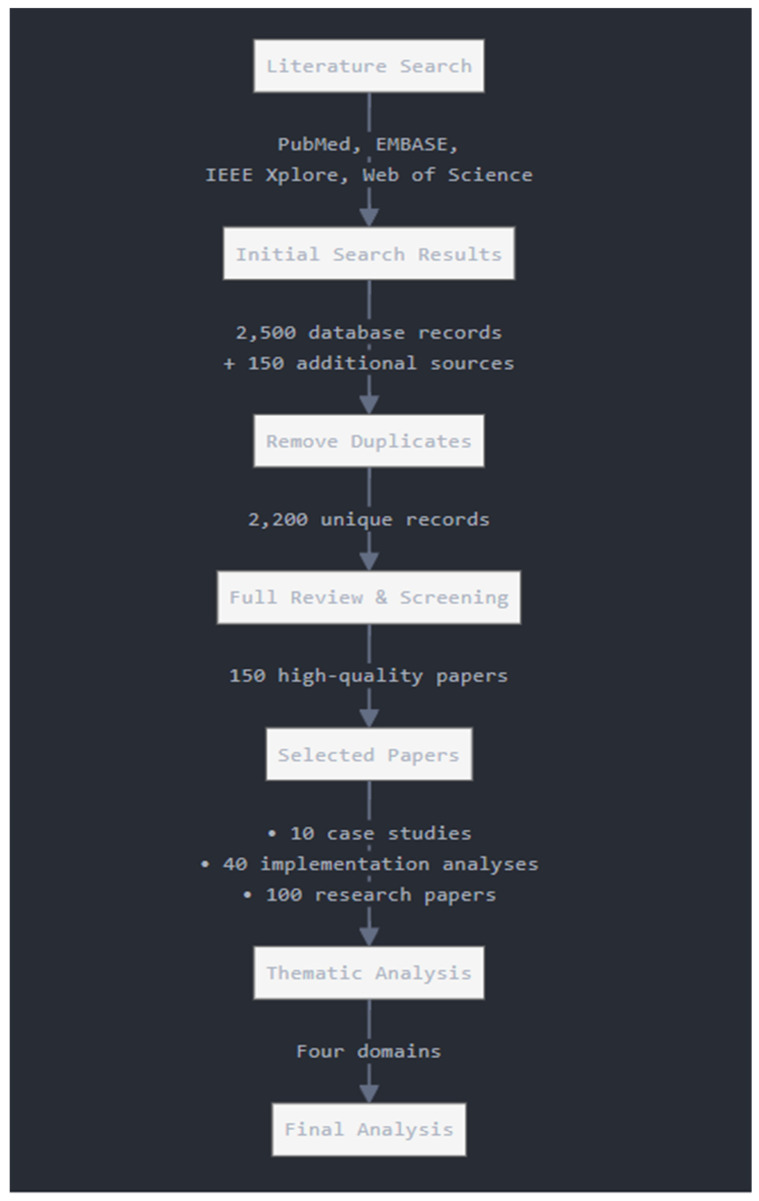
Literature search approach.

**Table 1 pharmacy-13-00041-t001:** Artificial intelligence/machine learning applications in pharmaceutical research—case studies.

Research Institute	Disease/Target	ML/AI Approach	Outcomes
**Atomwise [23,28]**	**Ebola**	**Deep learning algorithms for predicting binding affinity**	**Identified molecular sequences for Ebola treatment**
**Insilico Medicine [29,30]**	**Fibrosis**	**Use GANs (generative adversarial networks) for generating novel compounds**	**Generated novel compounds with significant activity against fibrosis target**
**Novartis and Atomwise [31]**	**Malaria and tuberculosis**	**AI algorithms for prioritising compounds based on predicted efficacy and safety profiles**	**Expedited identification of promising candidates and minimising resources spent on less viable options**
**Pfizer [32]**	**Breast cancer**	**ML algorithms for predicting compounds efficacy and safety**	**Identified potential breast cancer treatments with improved efficacy and safety profile**
**IBM and Pfizer [33]**	**Neurodegenerative diseases**	**AI-powered platform for identifying potential therapeutic targets**	**Identified novel targets for neurodegenerative diseases, including Alzheimer’s and Parkinsonism**
**Google and Stanford University [34,35]**	**Oncology/malignancies**	**Deep learning algorithms for analysing genomic data and identifying potential therapeutic targets**	**Identified potential therapeutic targets for various types of malignancies**
**Merck & Co. [36,37]**	**Cardiovascular diseases**	**ML algorithms for predicting compound efficacy and safety**	**Identified potential cardiovascular disease treatments with improved efficacy and safety profiles**
**AstraZeneca [38,39]**	**Respiratory diseases**	**AI-powered platform for identifying potential therapeutic targets**	**Identified novel targets for respiratory diseases including asthma and COPD**
**Sanofi [40]**	**Diabetes**	**ML algorithms for predicting compound efficacy and safety**	**Identified potential diabetes treatments with improved efficacy and safety profiles**
**Biogen [41,42]**	**Multiple sclerosis**	**AI-powered platform for identifying potential therapeutic targets**	**Identified novel targets for multiple sclerosis, including potential treatments for disease progression**

**Table 2 pharmacy-13-00041-t002:** Summary of cases illustrated above regarding artificial intelligence/machine learning applications in pharmaceutical research in professional and academic sectors.

Institute	AI/ML Applications	Primary Outcomes	Cost Savings
**Cleveland Clinic**	**Medication therapy management**	**42% reduction in readmission, 35% improved adherence, 58% better drug interaction detection**	**2.8M annually**
**Mayo Clinic**	**Antibiotic stewardship**	**45% reduction in inappropriate prescriptions, 30% decrease in *C. difficile* infections, 25% reduction in resistance rates**	**Not reported**
**Walgreens**	**Patient engagement system**	**40% increased adherence, 55% reduction in missing refills, 62% improved patient satisfactions**	**3.2M annually**
**Singapore General Hospital**	**Automated pharmacy system**	**75% fewer dispensing errors, 60% faster preparation, 45% improved staff productivity**	**1.5M annually**
**John Hopkins**	**ADR prediction**	**65% better ADR detection, 48% reduction in adverse events, 35% fewer emergency department visits**	**4.2M annually**
**NHS (UK)**	**Inventory management**	**55% fewer stock-outs, 40% reduced holding costs, 70% improved turnover**	**2.3M annually**
**Memorial Sloan Kettering**	**Oncology decision support**	**80% fewer preparation errors, 45% improved workflow, 50% faster verification**	**Not reported**
**Australian Pharmacy Networks**	**Triage system**	**50% reduced wait times, 65% better referrals, 40% increased service use**	**Not reported**
**Boston Children’s Hospital**	**Paediatric Medication Management**	**70% fewer dosing errors, 55% better dose adjustment, 45% fewer adverse effects**	**Not reported**
**UCFS Medical Center**	**Medication Reconciliation**	**65% fewer discrepancies, 50% improved accuracy, 40% time reduction**	**Not reported**

## Data Availability

Data are contained within the article.

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
