# Peer review of "Clinical and Operational Applications of Artificial Intelligence and Machine Learning in Pharmacy: A Narrative Review of Real-World Applications"

_pharmacy, 2025, doi:10.3390/pharmacy13020041_

Round 1

Reviewer 1 Report

Comments and Suggestions for Authors

This manuscript is an important narrative review that regroups the main areas where pharmaceutical care is provided, highlighting the main applications of artificial intelligence, analyzing benefits and future perspectives. The article is well written and well organized, however I suggest some amendments to get it published.

The introduction needs to be made more concise. The references are too few (five) and old, they should be updated to the last two years if possible.

Methodology: It would be useful to add a figure with a flow chart of the excluded and included studies.

Case study 2: Please delve into the concept of AI and AMR. How can the former counter the increasingly relevant issue of the latter?

Future Outlook: How AI can prevent and help address drug shortages? This issue need to be deepened.

Moreover, there is currently no internationally harmonized regulatory framework governing AI in healthcare. Please add this as a crucial aspect to be developed by public decision makers.

Thank you for giving me the opportunity to review this manuscript.

Reviewer 2 Report

Comments and Suggestions for Authors

I could detect that all the cases were generated using AI

The references are not structured as a review should and seem to be a collection of newspapers and articles.

The attributes of a scientific review are not met

Reviewer 3 Report

Comments and Suggestions for Authors

Dear authors,

I have read your paper and have the following comments:

MAJOR

-ML is a subset of AI. The mentioning of both, like in the title, suggest that these are two separate things with no overlap. Please correct throughout the document. 

-It would be good to include in the introduction a short description/explanation of the models that you refer to and their intended scope. Not everyone is a specialist

-All in all, the paper is difficult to read and the key points are not underlined sufficiently in my view. I suggest that the paper is rewritten to be more concise.

-please include the clear search strategy string

-not clear what "The review prioritized studies that demonstrated practical AI/ML implementations in pharmacy settings, provided quantifiable outcomes, and detailed implementation methodologies" means

-please keep references to companies to a minimum.

-how were the case studies selected and why are they considered the most relevant?

Minor

-please rewrite the introduction to avoid repetition and unnecessary text. Words like revolutionary should be avoided.

Reviewer 4 Report

Comments and Suggestions for Authors

General comments

The authors conducted a review to explore the application and implications of AI and machine learning within the context of pharmacy. The manuscript is well written and very interesting. However, the authors focused more on the positives of IA and machine learning and only briefly talked about their limitations. The authors need to expand what they wrote about the limitations of IA and machine learning and include other limitations such as job loss, privacy of patients, biases that may arise from how the algorithms were trained, ethical concerns, misuse of the algorithms etc…

Specific comments for revision:

a)      Major

    • The authors need to expand what they wrote about the limitations of IA and machine learning and include others such as job loss, privacy of patients, biases that may arise from how the algorithms were trained, ethical concerns, misuse of the algorithms etc.

b)      Minor

    • Case study 2: Did the authors mean “costs” in “contributed to a 30% decrease in healthcare-associated C”? is “C” an abbreviation?
    • Following up on the item above, the sentence that follows may need to be rewritten and clarified.

Round 2

Reviewer 1 Report

Comments and Suggestions for Authors

The authors have correctly addressed all comments and suggestions to improve the manuscript. In this form it can be accepted for publication. Thank you.

Reviewer 3 Report

Comments and Suggestions for Authors

No further comments.